# Decreasing trend of imported malaria cases but increasing influx of mixed *P. falciparum* and *P. vivax* infections in malaria-free Kuwait

**Jamshaid Iqbal** [ID]*, **Mohammad Al-Awadhi, Suhail Ahmad** [ID]

Department of Microbiology, Faculty of Medicine, Kuwait University, Kuwait, Kuwait

* jamshaid.rafique@ku.edu.kw

## Abstract

Malaria still continues to be the most important parasitic disease worldwide, affecting 228 million people and causing 405,000 deaths each year. In this retrospective study during 2013 to 2018, we documented the incidence of imported malaria infection and evaluated the impact of malaria preventive measures in Kuwait, a non-endemic country. The epidemiologic and demographic data of all malaria cases was collected from the Infectious Diseases Hospital, Kuwait where all suspected cases of malaria are referred for confirmation and therapeutic intervention. The diagnosis of malaria infection was done by microscopy of Giemsa stained blood films. Selected samples were retested with BinaxNOW® Malaria rapid test and molecular assay to reconfirm the *Plasmodium* spp. or mixed infection. Overall, 1913 (25.9%) malaria cases were detected, 81.5% of which were among male subjects. Male subjects had higher incidence of *P. vivax* malaria (113; 91.1%) and mixed infection with *P. falciparum* and *P. vivax* (1245; 90.0%) compared to females who had higher rate of *P. falciparum* infection (52.4%). An overwhelming majority of malaria cases (1895; 99.1%) were detected among expatriates from malaria-endemic countries; India (1012; 52.9%), Pakistan (390; 20.4%), Afghanistan (94; 4.9%) and African countries (313; 16.3%). Only 18 cases involved Kuwaiti nationals, all with a history of travel to African countries. The majority of malaria cases were detected during the summer and fall months (May-October). Our data showed that the incidence rate of imported malaria cases was stable during 2013 to 2018, however, the incidence of total malaria cases showed a declining trend over the years. This study confirms that the preventive program has been successful in reducing the incidence of imported malaria infections in Kuwait. The most striking finding of this study was high incidence of mixed infection with *P. falciparum* and *P. vivax*, with almost all (97%) cases among workers from India.

## Introduction

Despite widespread control and elimination efforts, malaria still continues to be the most important parasitic disease worldwide, affecting 228 million people and causing 405,000 deaths

**Data Availability Statement:** All relevant data are within the manuscript and its Supporting information files.

**Funding:** The financial support for this study was provided by the Research Sector and College of Graduate Studies, Kuwait University, Kuwait under the Research Project YM 06/14. The funder had no role in study design, data collection and analysis, decision to publish, or preparation of the manuscript.

**Competing interests:** The authors have declared that no competing interests exist.

in 2018, mostly in tropical and sub-tropical countries [1]. Although the major burden of malaria is in Africa, Asian countries suffer as well. Approximately 85% of malaria deaths occurred in 20 sub-Saharan African countries and India [1]. It is now well documented that population movements play an important role in the spread and introduction of malaria in nonendemic areas [2–5]. Over 40 million individuals from developed countries visit malaria-endemic countries each year, resulting in nearly 20,000 malaria cases in returning travelers in industrialized countries annually [4, 5]. Similarly, over 35 million people from malaria-endemic countries visit the developed world. Furthermore, military conflicts, civil unrest and ecologic changes have also contributed to global resurgence of malaria as large number of unprotected and nonimmune refugees have moved into malaria-endemic areas [6].

There have been no reports of local malaria transmission in most of the Middle eastern countries including Kuwait, except few reports from Saudi Arabia and Oman [7–12]. However, imported malaria was a major public health problem during the 1990s in Kuwait as >700 cases were detected annually among migrant workers from malaria-endemic countries [12]. The current population of 4.7 million inhabitants in the State of Kuwait comprises 70% expatriate workers and their family members originating mainly from South Asian and African countries [13]. Although total number of malaria cases in Kuwait initially decreased over the years as a result of malaria control program initiated in 1985 [12], no data are available to confirm this decreasing trend in recent years. This retrospective epidemiologic study was carried out to reevaluate the impact of ongoing preventive measures on the current status of imported malaria cases during 2013 to 2018 in Kuwait.

## Materials and methods

### Extraction, categorization and entry of epidemiological data

Malaria infection is a notifiable disease in Kuwait. Though all migrant workers from malaria endemic countries entering Kuwait for the 1st time are required to carry a recent malaria-free certification issued by the designated Diagnostic Laboratories in their home countries, however, they are also screened for malaria infection at the Ports and Borders Health Centers in Kuwait. Kuwait is a small country consisting of six administrative governorates. All malaria positive cases are then referred to the Malaria Reference Laboratory, Infectious Diseases Hospital (MRL, IDH), Kuwait for treatment and follow up. In addition, any resident in Kuwait suspected of malaria infection is referred to the MLR, IDH for confirmation and therapeutic management. For all cases, information on civil identification number, age, gender, nationality, referring hospital, date of sample collection, major clinical symptoms, recent travel history and test results are recorded in the MLR, IDH registry logbooks.

The epidemiologic and demographic data of all cases was collected from the registry logbooks of MRL, IDH during the period January 2013 to December 2018.

### Diagnosis of malaria

The diagnosis of *Plasmodium* spp. infection was performed on the venous whole blood collected in EDTA collection tubes. A thick and thin blood film were prepared from each specimen, fixed in ethanol and stained with Giemsa solution for 30 min to determine the parasite stage by light microscopy. Serological testing was performed on blood samples to confirm *Plasmodium* spp. by using the rapid BinaxNOW® Malaria rapid diagnostic test (Alere, Cologne, Germany) according to manufacturer's instructions and as described previously [14]. Briefly, 15 μl of whole blood was added to the designated purple area of nitrocellulose test strip labeled with anti-HRP-2 and aldolase monoclonal antibodies. Subsequently, Reagent A (tris buffer + Triton® X-100 + sodium azide) was used to process the test and the results were read after

15 min from the result window. A positive test result showed both the control line and test line (s), while a negative test result produced only the control line. Absence of control line indicated an invalid result.

## Detection of *Plasmodium* species by real-time polymerase chain reaction (PCR)

Specimens not processed for PCR within 2 days were stored at 2–8˚C. A genus-specific primer set corresponding to the 18S rRNA was used to amplify the target sequence. Real-time PCR detection of *Plasmodium* species was performed blinded to the microscopy and malaria rapid test results. All PCRs were performed using the LightCycler-FastStart DNA Master Hybridization Probes kit (Roche Applied Science) as described earlier [15]. An internal control was used to monitor the DNA isolation procedure and to check for possible PCR inhibition. Briefly, each reaction had a final concentration of 4 μM MgCl2, 0.5 μM primer PF1, 1.0 μM primer PF2, 0.2 μM probe PF3, and 0.4 μM probe PF4. The amplification and detection was carried out as described earlier with minor modifications to the thermal profile [15]. Amplification of different *Plasmodium* species yielded products with different melting temperature curves and mixed infections showed double curve with appropriate species temperatures. The limit of detection (LOD) of the assay was determined at 1–5 parasites/μl of blood ($\leq$ 0.0001% of infected red blood cells).

## Statistical analysis

Descriptive statistical analysis was performed, and Pearson's Chi-square was used to test association of infection with patient's sociodemographic profile. Student's t-test was used to measure the difference in age between infected and non-infected patients. A *P*-value less than 0.05 was considered as statistically significant.

## Ethical clearance

This study was approved by the Health Sciences Center Ethical Committee, Kuwait University, and the Ethical Committee for the Protection of Human Subjects in Research, Ministry of Health, Kuwait, under reference no. 187/2014 which is in agreement with the code of ethics of the World Medical Association. Personal details about individual participants were not disclosed and the data were analyzed anonymously.

## Results

Between January 2013 and December 2018, a total of 7386 subjects were referred to the Infectious Diseases Hospital for malaria investigation. Overall, 1913 (25.9%) patients tested positive for malaria. Most malaria cases were detected among male patients (*n* = 1560; 81.5%) (Table 1). However, the number of female subjects infected with *P. falciparum* was significantly higher (*n* = 189; 52.4%) in comparison with male patients (*n* = 172; 47.6%), whereas male subjects had a significantly higher incidence of *P. vivax* malaria (*n* = 113; 91.1%) and mixed infection with *P. falciparum* and *P. vivax* (*n* = 1245; 90.0%) in comparison with female subjects (*n* = 11; 8.9% and *n* = 138; 10.0%, respectively) (*P* <0.001) (Table 1).

The age of subjects with malaria ranged from 1 to 73 years with a mean of 34.1 years (95% CI: 33.6, 34.5). The mean age of female subjects [30.5 years (95% CI: 29.6, 31.4)] was significantly lower than that of male subjects [34.9 years (95% CI: 34.4, 35.4)] (*P* <0.001) (Table 1). In particular, the mean age of females [27.1 years (95% CI: 26.2, 28.0)] infected with *P. falciparum* was significantly lower in comparison with that of *P. falciparum*-infected male patients

**Table 1. Distribution of malaria cases by causative *Plasmodium* spp., gender and mean age in Kuwait (2013–2018).**

| *Plasmodium* spp. | n, (%) | Mean age in years (95% Confidence Interval) | P-value |
|---|---|---|---|
| **All *Plasmodium* spp.** | 1913 (100.0%) | 34.1 (33.6, 34.6) | |
| Male | 1560 (81.5%) | 34.9 (34.4, 35.4) | <**0.001** |
| Female | 353 (18.5%) | 30.5 (29.5, 31.5)[b] | |
| ***P. falciparum* + *P. vivax*** | 1383 (72.3%) | 35.2 (34.7, 35.7) | |
| Male | 1245 (90.0%) [a] | 35.2 (34.6, 35.8) | <**0.001** |
| Female | 138 (10.0%) | 35.1 (33.2, 37.0) | |
| ***P. falciparum*** | 361 (18.9%) | 30.2 (29.2, 31.2) | |
| Male | 172 (47.6%) | 33.6 (31.9, 35.3) | <**0.001** |
| Female [a] | 189 (52.4%)[a] | 27.1 (26.2, 28.0) [b] | |
| ***P. vivax*** | 124 (6.5%) | 35.4 (33.7, 37.1) | |
| Male | 113 (91.1%) [a] | 35.6 (33.9, 37.3) | <**0.001** |
| Female | 11 (8.9%) | 33.0 (27.3, 38.7) | |
| ***P. falciparum* + *P. ovale*** | 25 (1.3%) | 29.4 (25.8, 33.0) | |
| Male | 20 (80.0%) | 29.2 (25.0, 33.4) | **N/A** |
| Female | 5 (20.0%) | 30.4 (20.8, 40.0) | |
| ***P. falciparum* + *P. malariae*** | 12 (0.6%) | 27.9 (22.7, 33.1) | |
| Male | 5 (41.7%) | 28.0 (15.2, 40.8) | **N/A** |
| Female | 7 (58.3%) | 27.9 (21.2, 34.6) | |
| ***P. ovale*** | 8 (0.4%) | 30.6 (16.7, 44.5) | |
| Male | 5 (62.5%) | 29.0 (7.6, 50.4) | **N/A** |
| Female | 3 (37.5%) | 33.3 (N/A) | |

[a] Indicates points of significance based on gender.

[b] Indicates points of significance based on mean age (years).

[33.6 years (95% CI: 31.9, 35.3)] (*P* <0.001). Both the total number of malaria and *P. falciparum* infections among the male and female cases showed a single peak of age distribution on histograms, 22–33 *vs* 20-28-year age group for total malaria cases & 21–34 *vs* 23-26-year age group for *P. falciparum* cases respectively (S1 Fig in S1 File).

Only 18 (0.9%) positive cases were among Kuwaiti nationals returning from overseas visit while the remaining cases had come to Kuwait from several other countries in the descending order as follows: India (1012; 52.9%), Pakistan (390; 20.4%), Afghanistan (94; 4.9%), Cameroon (54; 2.8%), Sudan (48; 2.5%), Ghana (46; 2.4%), other African countries (165; 8.6%) and other countries (86; 4.4%). Thus a large majority (1507; 78.8%) were non-Arab Asians mainly originating from India, Pakistan and Afghanistan, while malaria cases originating from non-Arab African countries (303; 15.8%) and Arab countries (82; 4.3%) formed minority group (Fig 1).

Surprisingly, the majority (1420 of 1913, 74.2%) of malaria cases had a mixed infection with both *P. falciparum* and *P. vivax* (1383; 72.3%) or *P. falciparum* and *P. ovale* (25; 1.3%) or *P. falciparum* and *P. malariae* (12; 0.6%) while other cases were caused by *P. falciparum* alone (361; 18.9%) or *P. vivax* (124; 6.5%) alone (Fig 2). Almost all malaria cases which had a mixed infection with *P. falciparum* and *P. vivax* (1303; 97.0%) originated from India (878; 67.4%), Pakistan (341; 26.2%) and Afghanistan (84; 6.4%).

The annual number of reported malaria cases remained stable between 262–312 cases during 2013–2015 but showed a significant increase from 2015 to 2017 (409 cases) before declining again in 2018 (Fig 3). The annual number of cases which had mixed infection with *P.*

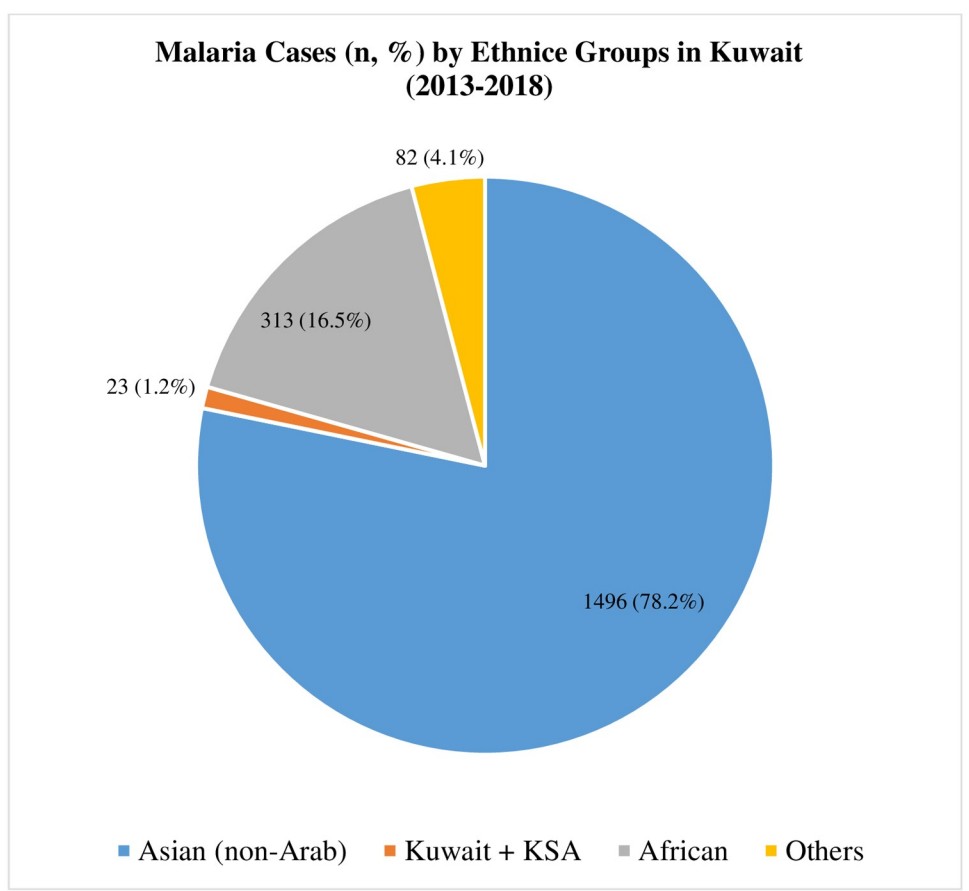

**Fig 1. A pie chart showing the share (n, %) of reported malaria cases among the ethnic groups in Kuwait (2013–2018).**

*falciparum* and *P. vivax* showed a declining trend during 2013 to 2015 but showed an increasing trend during 2015 to 2017 before declining again in 2018. On the other hand, the data showed an increasing trend in the number of *P. falciparum* infections alone and a decreasing trend in the number of *P. vivax* cases alone from 2013 to 2018 (Fig 3). The number of malaria cases detected in Kuwait were significantly higher during the summer and fall months with maximum number of cases detected during the summer months of July (252; 13.2%), August (273; 14.3%) and September (265, 13.9%) (S2 Fig in S1 File).

## Discussion

The ever increasing travel for business and/or leisure and migratory movements for employment or displacement/forced movement of refugees within and across national borders due to geopolitical conflicts have changed epidemiological characteristics of imported malaria in many non-malaria-endemic countries [4–6, 16]. The relative risk of malaria infection in travelers visiting different countries can be roughly estimated by using the surveillance data and the disease characteristics in different areas. Approximately, 3.3 million of ~4.7 million inhabitants in 2019 in Kuwait were expatriate workers from many malaria-endemic countries. A systematic survey of imported malaria cases in Kuwait between January 2013 and December 2018 was conducted and the data were compared with a previous survey conducted nearly a decade ago.

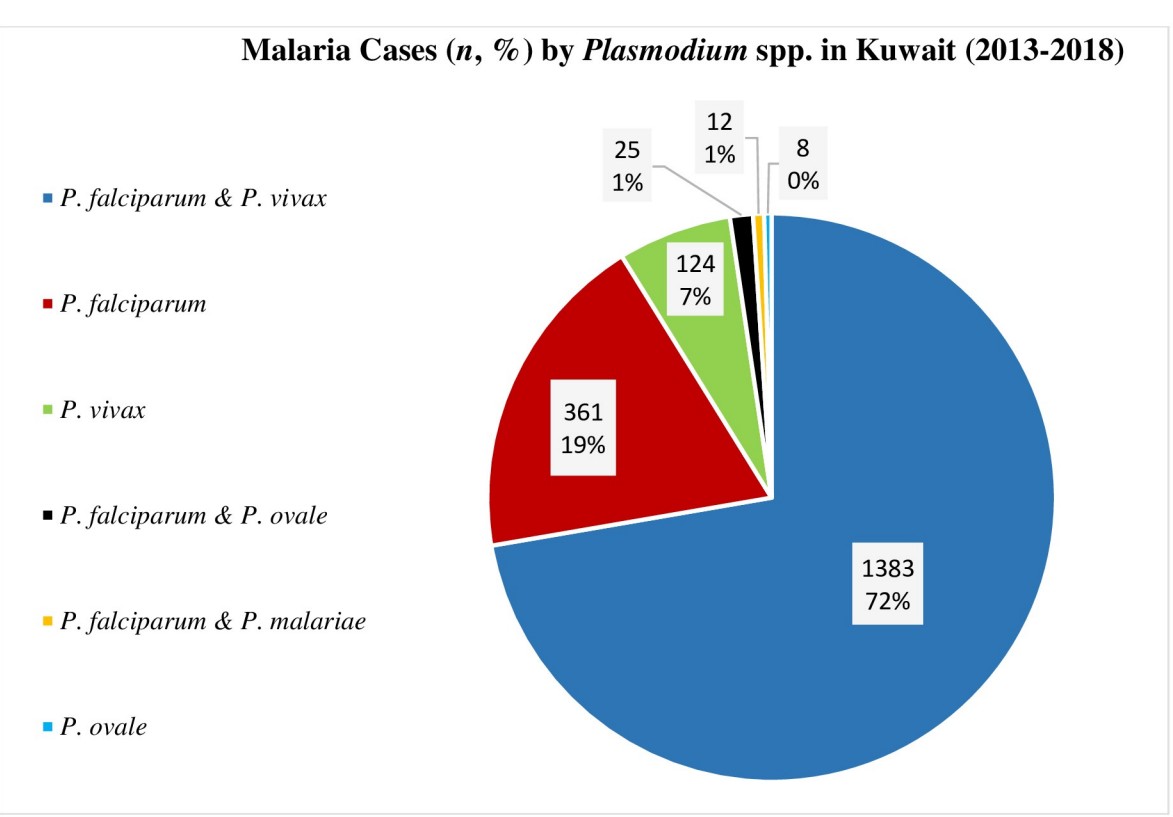

**Fig 2. A pie chart showing the share (n, %) of reported malaria cases infected with different species of malaria in Kuwait (2013–2018), mixed infection with *P. falciparum* and *P. vivax* constituting the major infection type.**

Our data showed that 1913 malaria cases were detected among 7386 suspected cases in Kuwait with yearly low of 262 cases in 2014 to a high of 409 cases in 2017. This rate is considerably lower than the data reported in an earlier study conducted during 1985–2000 showing an infection rate of >1200 cases/year of imported malaria cases [12]. Since the total population of Kuwait has changed drastically in recent years due to influx of large number of expatriates mainly from malaria-endemic countries, it was of interest to determine the incidence rate of malaria during this study period. The data showed that the incidence rate was stable during 2013 to 2018 (Fig 4). When the data were combined with data from our previous study [12], it became apparent that the incidence of total malaria cases have shown a declining trend in the number of imported malaria cases over the years, especially after 1994 (Fig 4). This is likely attributed to the well-organized malaria control program initiated by the Preventive Department of the Ministry of Health in 1996, comprising pre-screening of migrant workers from malaria-endemic countries in their home countries before their travel to Kuwait, further screening of all expatriates upon entry into Kuwait, spraying of insecticides in agricultural lands and mandatory reporting of positive malaria cases. On the contrary, number of malaria cases reported among returning resident expatriates increased after visiting their home country, likely due to their semi-immune status. Similar observations have also been made in other adjoining GCC countries including Bahrain [17], Qatar [18, 19], and Saudi Arabia [20–22].

The clinical and epidemiological features recorded in our study showed that an overwhelming majority of malaria cases (1895; 99.1%) were detected among expatriates (imported malaria cases), originating from malaria-endemic countries, particularly from India, Pakistan and Afghanistan, as expected. Only 18 cases involved Kuwaiti nationals, all with a history of

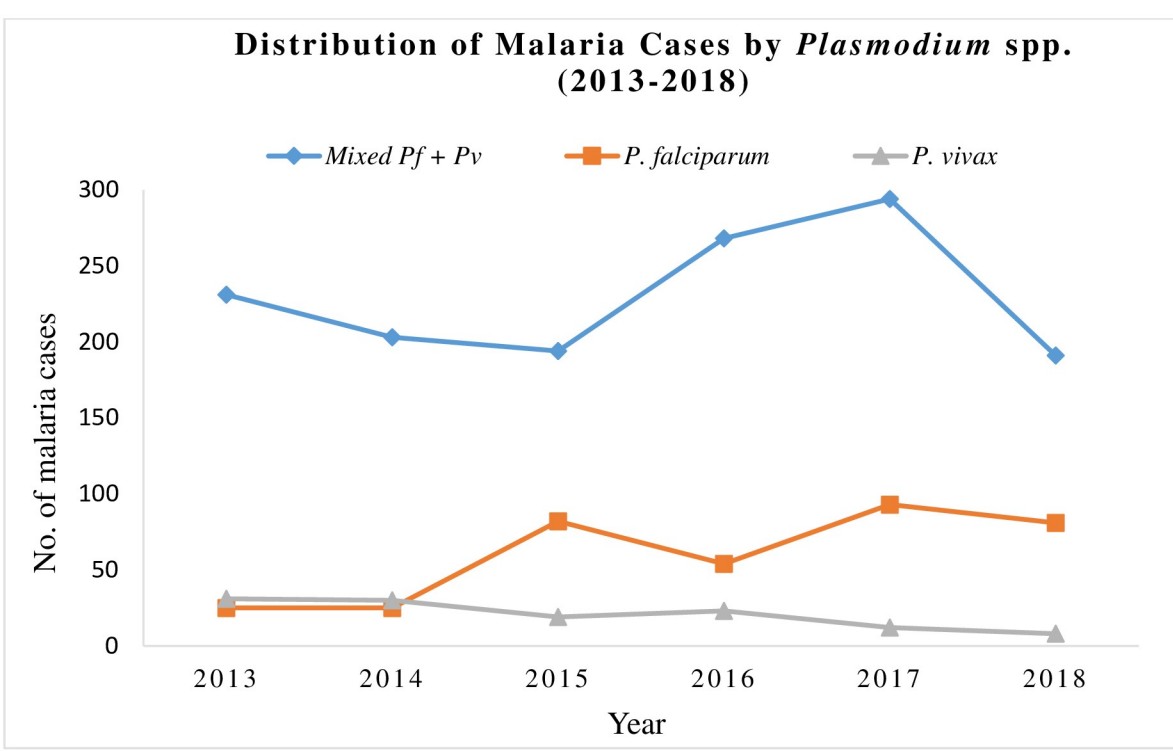

**Fig 3. A line graph showing the annual number of reported malaria cases by *Plasmodium* spp. with significantly higher numbers of *P. falciparum* and *P. vivax* mixed infection in Kuwait (2013–2018) (P <0.001).**

travel to malaria-endemic African countries. The highest number of malaria cases (1012; 52.9%) were detected among Indians which also form the biggest single ethnic group among expatriate residents in Kuwait [13]. Similar data have also been reported from other Arabian Gulf countries, such as Qatar, Bahrain, United Arab Emirates and Saudi Arabia which also have large expatriate population, particularly from the Indian sub-continent [17–19, 23, 24].

The majority of malaria cases were detected among male subjects and during the summer and fall months (May-October), which coincides with the rainy season/peak malaria infection period in the home countries of expatriate workers who either came or returned to Kuwait after summer holidays. Males also comprised >90% of all imported malaria cases in Qatar [19], UAE [23] and Saudi Arabia [24]. Furthermore, similar seasonal trend was also reported from other GCC countries with most imported cases detected during the months of August-September [18, 23, 25]. The malaria cases among returning expatriate subjects could be the result of declined immunity due to their prolonged stay in non-endemic Kuwait which probably resulted in increased susceptibility to malaria infection during their subsequent visits to home countries. Although there is no sensitive biomarker to measure the anti-parasite immunity in the residents or travelers, current evidence indicates variable responses to malaria infection depending on the anti-parasite immune status of the host [4].

A striking observation of our study was the detection of mixed *P. falciparum* and *P. vivax* infections in the majority (1383; 72.3%) of malaria cases followed by *P. falciparum* monoinfection in 361 (18.9%) cases. Nearly all (97%) of the mixed infection cases were found among subjects from India. The diagnosis was made by using a highly sensitive serological test [14]. Misdiagnosis was ruled out by retesting several selected samples by a molecular (PCR) assay [15]. In an earlier survey carried out during 1990–2000 in Kuwait, *P. vivax* was the dominant

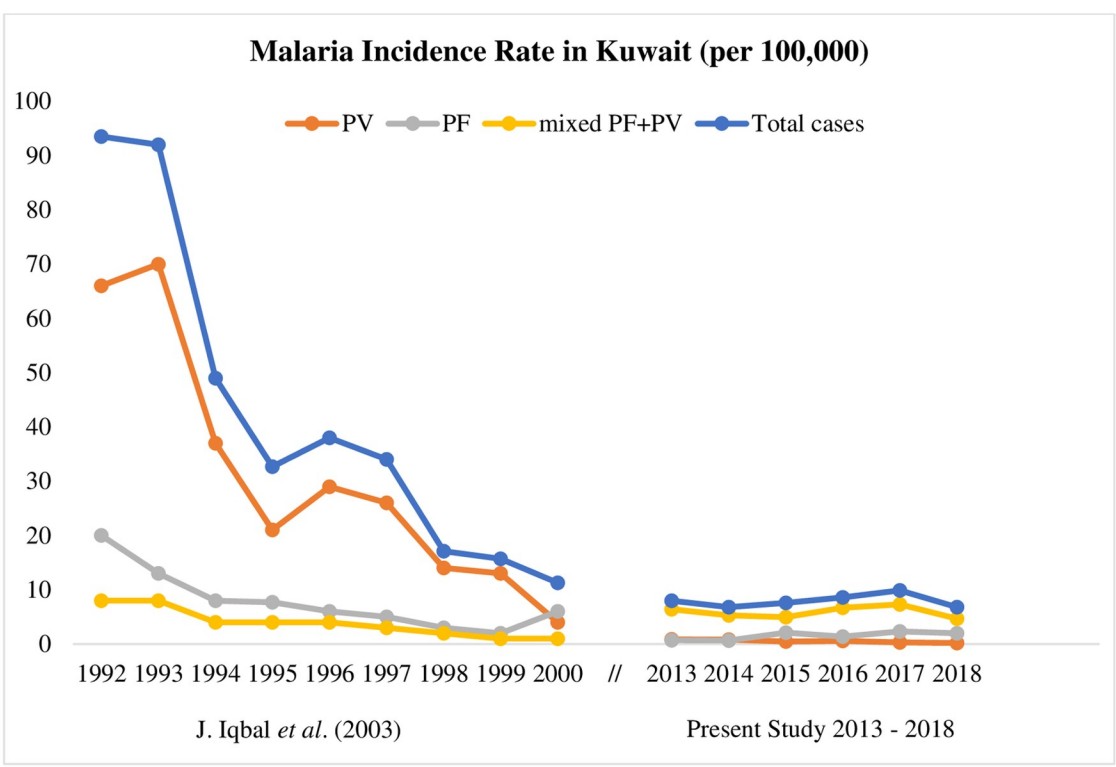

**Fig 4. A line graph showing a comparison of reported malaria incidence rates in Kuwait (per 100,000 persons) from a previous study (1992–2000) [12] and this study (2013–2018).**

*Plasmodium* spp. detected among the migrant workers, including expatriates from India [12]. It is pertinent to mention here that *P. falciparum* which had dominated India's malaria cases previously, is now showing a decreasing trend over the last few years, 65.4% in 2017 to 46.4% in 2019 while majority of malaria cases are now caused by *P. vivax* [26]. Consistent with our findings, several recent studies from India which had employed molecular (PCR) methods for detection of malaria cases had also reported mixed *P. falciparum/P. vivax* infections in 11% to 45% of patient samples [27, 28]. Some studies have also concluded that since field diagnosis of malaria at the primary health care level in India is mostly performed by microscopy which has low sensitivity/accuracy in diagnosing multiple species malaria, cases of mixed species infection are usually missed [27, 29–31]. This is also evident from the fact that ~17% of mixed infections were initially identified as monoinfections due to *P. falciparum* in one study from India [29]. A recently developed field-applicable, rapid, ultrasensitive diagnostic assay that specifically detects DNA sequences from all *Plasmodium* species in symptomatic and asymptomatic malaria is expected to detect low level and mixed-*Plasmodium* spp. infections more efficiently [32]. The new malaria diagnostic method combines an optimized 10-minute rapid sample preparation protocol with the CRISPR-based SHERLOCK system to enable highly specific and sensitive Plasmodium detection in another 60 minutes in simple reporter devices. It is published in *PNAS*.

The detection of mixed *P. falciparum* and *P. vivax* infections in 72.3% of malaria cases in Kuwait is the highest reported so far from among the GCC countries. The majority of imported malaria cases in Qatar and United Arab Emirates are caused by *P. vivax* followed by *P. falciparum* [19, 23], and mixed *P. falciparum* and *P. vivax* infections are less frequent, occurring only in 2% of the cases [19, 33]. On the contrary, *P. falciparum* malaria is more common

in locally acquired malaria cases while an increasing proportion of imported cases were caused by *P. vivax* in Saudi Arabia [7, 20, 24]. Mixed *P. falciparum* and *P. vivax* infections occurred in only ~7% of malaria cases in one study [8], while in another study, nearly 2% of PCR-confirmed malaria cases (n = 369) were due to mixed infections that were missed by microscopic analysis alone [34].

All mixed infections with *P. falciparum* + *P. vivax* were treated with artemether+lumefantrine, given twice a day for 3 days. This treatment was followed by treatment with the standard dose of primaquine (0.25 mg/kg/day) for 28 days to treat hypnozoite stage of the *P. vivax*. Though the WHO has also recommended a higher dosage of primaquine (0.5 mg/kg/day) for 14 days, however, we used the standard dose for primaquine to minimize the side effect of hemolysis especially in patients with glucose-6-phosphate dehydrogenase (G6PD) deficiency. However, up to 20% of the mixed *Plasmodium* cases showed relapse of mixed *Plasmodium* infection 3–4 months after therapeutic/radical anti-malarial treatment with no travel history to the Indian sub-continent. A number of studies have reported in mixed malaria, an increase in relapse rates by *P. vivax* following treatment of *P. falciparum* [35]. Though, the most common cause of *P. vivax* relapse is poor adherence to treatment or inappropriate dose, however, failure in treatment due to impairment in CYP2D6 enzyme is also being investigated [36].

Studies are urgently needed to investigate the molecular basis of resistance of mixed *Plasmodium* infections in Kuwait and correlation with disease severity and outcome. Recently, a number of reports have documented an influx of diverse, drug resistant malaria cases in the neighboring countries of Qatar and Saudi Arabia [10, 33, 37]. Moreover, the ICMR-National Institute of Malaria Research (ICMR-NIMR) in India has recently reported drug resistant strains of *P. falciparum*, mainly from East and Northeast of India [17, 38].

Although autochthonous malaria cases have been reported from two GCC countries (Oman and Saudi Arabia) [7–9, 21], no autochthonous malaria cases have been detected in Kuwait so far. However, certain risk factors have emerged recently in and around the country which may endanger the present situation. These factors are: i) A change in the present ecological and meterological conditions due to an enthusiastic drive for making Kuwait green and extending plantations throughout country. Presently the larval density levels of *A. stephensi* and *A. pulcherrimus* are well below the threshold for transmission due to arid conditions. ii) the extensive use of water for agricultural irrigation and domestic purposes, iii) recent reports of imported drug resistant malaria cases in the neighboring country, Qatar [33]. A significant change in the drug sensitivity pattern of *P. falciparum* isolates from these countries and this report of imported mixed *Plasmodium* infections from India will have an important impact on the future therapeutic, as well as prophylactic policies for *P. falciparum* infection in Kuwait.

In summary, this study confirms that the proactive preventive program has been successful in reducing the incidence of imported malaria infections in Kuwait and has now shifted the burden of malaria cases towards returning expatriate residents, who are now regarded as the high-risk group for acquisition of travel-associated malaria. This group is now being targeted for prevention strategies such as education and information on various preventive measures before their travel to endemic countries. The most striking finding of this study was prevalence of mixed infections with both *P. falciparum* and *P. vivax* (72.3%) with almost all (97%) cases involving migrant workers from India. This observation needs to be followed up with more detailed comparative analysis of various diagnostic tests at the Malaria Center, Infectious Disease Hospital, Kuwait and recording of comprehensive demographic data, especially the location/area of residence of the incoming expatriate population to follow up the recent update of malaria incidence infection of that location. In addition, up to 20% of the mixed *Plasmodium* cases showed recrudescence of mixed *Plasmodium* infection 3–4 months after therapeutic and radical anti-malarial treatment. A significant change in the drug sensitivity pattern of *P.*

*falciparum* isolates from these countries and the finding of imported mixed *Plasmodium* infections from India will have an important impact on the future therapeutic, as well as prophylactic policies for *P. falciparum* infections in Kuwait.

## Supporting information

**S1 File.**
(DOCX)

## Acknowledgments

We would like to thank Dr. Mamoun Al-Qaseer and Dr. Ali Sher for sharing their epidemiological data on malaria cases at the Malaria Reference Laboratory, Infection Diseases Hospital, Ministry of Health, Kuwait.

## Author Contributions

**Conceptualization:** Jamshaid Iqbal, Mohammad Al-Awadhi, Suhail Ahmad.

**Data curation:** Mohammad Al-Awadhi.

**Formal analysis:** Mohammad Al-Awadhi, Suhail Ahmad.

**Methodology:** Mohammad Al-Awadhi.

**Project administration:** Jamshaid Iqbal.

**Validation:** Jamshaid Iqbal, Suhail Ahmad.

**Writing – original draft:** Mohammad Al-Awadhi.

**Writing – review & editing:** Jamshaid Iqbal, Suhail Ahmad.

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
