## [Decision Letter · Decision Letter 0]

28 Oct 2020

PONE-D-20-30612

Decreasing trend of imported malaria cases but increasing influx of mixed P. falciparum and P. vivax infections in malaria-free Kuwait

PLOS ONE

Dear Dr. Iqbal,

Thank you for submitting your manuscript for review to PLoS ONE. After careful consideration, we feel that your manuscript will likely be suitable for publication if it is revised to address a specific topic raised by the reviewers. According to reviewer, there are some specific areas where further improvements would be of substantial benefit to the readers, including methods and results.   For your guidance, a copy of the reviewers' comments was included below. 

We look forward to receiving your revised manuscript.

Kind regards,

Luzia Helena Carvalho, Ph.D.

Academic Editor

PLOS ONE

Journal Requirements:

2. Please state whether the baseline demographic characteristics of the study populations were recorded. If so, please provide a table summarising these.

Reviewers' comments:

Reviewer's Responses to Questions

**Comments to the Author**

1. Is the manuscript technically sound, and do the data support the conclusions?

Reviewer #1: Yes

Reviewer #2: Yes

2. Has the statistical analysis been performed appropriately and rigorously? 

Reviewer #1: Yes

Reviewer #2: Yes

3. Have the authors made all data underlying the findings in their manuscript fully available?

Reviewer #1: Yes

Reviewer #2: Yes

4. Is the manuscript presented in an intelligible fashion and written in standard English?

Reviewer #1: Yes

Reviewer #2: Yes

5. Review Comments to the Author

Reviewer #1: The authors update the epidemiological situation in Kuwait, which basically depends on imported malaria, and shows that Kuwait has adequate policies and a health system that is currently preventing the emergence of autochthonous cases. The high proportion of mixed malaria P. vivax + P. falciparum is noteworthy, which the affected people acquired in their countries of origin, mainly India and which are not detected because they do not routinely use diagnostic methods with limited sensitivity. The significance of this finding is that mixed infection is much more frequent than is diagnosed in the countries of origin and the need in the near future to use radical university treatment for both species.

It would be convenient for the authors to specify the treatment scheme used in Kuwait, they mention the use of chloroquine plus primaquine (there is no dose information) for 28 days, however they observe 20% relapses. Assuming that in Kuwait the dose of primaquine (PQ) that they use is the one recommended by WHO (0.25 mg / kg), the total dose per kilo would be at high dose levels of PQ (7 mg / kg) where the frequency of relapses is very low (~ 1%). However, it is also published in mixed malaria, an increase in relapses by vivax after treatment of P. falciparum. It would justify the authors to make a comment on adherence to treatment in the 28-day period. I also suggest that the authors make some comment on the possibility of future use of tafenoquine due to the high relapse rate, Kuwait being a country where the economic factor is not a limitation.

Reviewer #2: Thank you very much for giving me the opportunity to review the manuscript “PONE-D-20-30612” titled as “Decreasing trend of imported malaria cases but increasing influx of mixed P. falciparum and P. vivax infections in malaria-free Kuwait”. The authors sought to characterize the incidence of imported malaria infection and evaluated the impact of malaria preventive measures in Kuwait using a retrospective chart review over a 6-year period. The authors note a total of 1913 (25.9%) malaria cases, 81.5% of which were among male subjects. Also, majority of malaria cases (1895; 99.1%) were detected among expatriates from 38 malaria-endemic countries. Overall, the paper is nicely written. However, there are spaces for improvement. I have few comments:

Methods:

• What prevention and control measures were in place?

• Data collection paragraph does not detail the procedure for data abstraction. There is need to describe how the facility collected malaria information. Who collected the samples? All these relevant information should be clearly indicated in the “Methods and materials" section.

• How was consistency assessed? It is important to include data with a few missing variables to avoid bias, please clarify how many records were excluded due to incomplete data.

• How malaria positive patients were administered?

Discussion:

• Reference for lines 252-253

• Please re-read for grammar edits pp. 12, line 255 insert comma (,) after “On the contrary,…”

Thank you very much.

Araya Gebresilassie (Ph.D)

Department of Zoological Sciences, Addis Ababa University

6. PLOS authors have the option to publish the peer review history of their article (what does this mean?). If published, this will include your full peer review and any attached files.

Reviewer #1: **Yes: **Alejandro Llanos-Cuentas MD, PhD

Reviewer #2: **Yes: **Araya Gebresilassie

---

## [Author Response · Author response to Decision Letter 0]

5 Nov 2020

Dated: 5th November, 2020

Dear Ms Carvalho

Re: PONE-D-20-30612

‘Decreasing trend of imported malaria in cases but increasing influx of mixed P. falciparum and P. vivax infections in malaria-free Kuwait’.

With reference to above manuscript, I take this opportunity to thank you and both the reviewers, Prof. Llanos-Cuentas and Prof. Gebresilassie on their thorough review and excellent suggestions to further improve the quality of the manuscript.

We have made the appropriate changes and edits as suggested by the Reviewers. The itemized responses to their queries are follows:

Both the reviewers have agreed and have no suggestion/comments to the following questions:

1. Is the manuscript technically sound, and do the data support the conclusions?

2. Has the statistical analysis been performed appropriately and rigorously? 

3. Have the authors made all data underlying the findings in their manuscript fully available?

4. Is the manuscript presented in an intelligible fashion and written in standard English? 

Review Comments to the Author

Reviewer 1:

Query 1 & 2: Specify the malaria treatment scheme used in Kuwait & comment on 28-day treatment with primaquine 

Answer: We have edited the text as follows on Page 13, lines 259-270 and included two additional references (#35, 36).

All mixed infections with P. falciparum + P. vivax were treated with artemether+lumefantrine, given twice a day for 3 days. This treatment was followed by treatment with the standard dose of primaquine (0.25 mg/kg/day) for 28 days to treat hypnozoite stage of the P. vivax. Though the WHO has also recommended a higher dosage of primaquine (0.5 mg/kg/day) for 14 days, however, we used the standard dose for primaquine to minimize the side effect of hemolysis especially in patients with glucose‐6‐phosphate dehydrogenase (G6PD) deficiency. However, up to 20% of the mixed Plasmodium cases showed relapse of mixed Plasmodium infection 3-4 months after therapeutic/radical anti-malarial treatment with no travel history to the Indian sub-continent. A number of studies have reported in mixed malaria, an increase in relapse rates by P. vivax following treatment of P. falciparum [35]. Though, the most common cause of P. vivax relapse is poor adherence to treatment or inappropriate dose, however, failure in treatment due to impairment in CYP2D6 enzyme is also being investigated [36]. 

Query #2: make some comment on the possibility of future use of tafenoquine

Answer: Though the use of Tafenoquine as a single-dose (300 mg) therapy for P. vivax relapse prevention was discussed but not yet approved by the Ministry of Health, Kuwait, considering its safety challenge in causing severe hemolytic anemia in people with G6PD deficiency. However, we hope that the Ministry of Health may approve its use soon considering rise of P. vivax relapses following treatment of mixed malaria infections.

Reviewer 2:

Query 1 & 2: What prevention and controlmeasures were in place & What was the detail on Data collection? 

Answer: We have edited the text as follows on Page 4, lines 74-86 under Materials & Methods.

Malaria infection is a notifiable disease in Kuwait. Though all migrant workers from malaria endemic countries entering Kuwait for the 1st time are required to carry a recent malaria-free certification issued by the designated Diagnostic Laboratories in their home countries, however, they are also screened for malaria infection at the Ports and Borders Health Centers in Kuwait. Kuwait is a small country consisting of six administrative governorates. All malaria positive cases are then referred to the Malaria Reference Laboratory, Infectious Diseases Hospital (MRL, IDH), Kuwait for treatment and follow up. In addition, any resident in Kuwait suspected of malaria infection is referred to the MLR, IDH for confirmation and therapeutic management. For all cases, information on civil identification number, age, gender, nationality, referring hospital, date of sample collection, major clinical symptoms, recent travel history and test results are recorded in the MLR, IDH registry logbooks. The epidemiologic and demographic data of all cases was collected from the registry logbooks of MRL, IDH during the period January 2013 to December 2018. 

Query 3: How was consistency assessed? 

Answer: As mentioned above, the epidemiological data for all referred cases was recorded in the MLR, IDH registry logbooks. There were just 4-5 cases with missing information on their nationality and/or travel history, which we did not include in our data as the number was very small.

Query 4: How malaria positive patients were administered?

Answer: Please refer to our answer to query # 1 & 2 for the 1st reviewer, and is the edited text is added at Page 13, lines 259-270 

Query 5: Reference for lines 252-253

Answer: The reference # 19 & 23 is included, as suggested.

Query 6: Please re-read for grammar edits pp. 12, line 255 insert comma (,) after “On the contrary,…”

Answer: The sentence is edited, as suggested.

I hereby, certify that all the authors have reviewed and agreed to the revised version and answers to reviewer’s queries. We hope that the revised version of the manuscript will be acceptable for publication to the Journal PLOS ONE.

Regards

Dr. Jamshaid Iqbal

Corresponding Author

---

## [Decision Letter · Decision Letter 1]

25 Nov 2020

Decreasing trend of imported malaria cases but increasing influx of mixed P. falciparum and P. vivax infections in malaria-free Kuwait

PONE-D-20-30612R1

Dear Dr. iqbal,

We’re pleased to inform you that your manuscript has been judged scientifically suitable for publication and will be formally accepted for publication once it meets all outstanding technical requirements.

Kind regards,

Luzia Helena Carvalho, Ph.D.

Academic Editor

PLOS ONE

Additional Editor Comments (optional):

Reviewers' comments:

Reviewer's Responses to Questions

**Comments to the Author**

1. If the authors have adequately addressed your comments raised in a previous round of review and you feel that this manuscript is now acceptable for publication, you may indicate that here to bypass the “Comments to the Author” section, enter your conflict of interest statement in the “Confidential to Editor” section, and submit your "Accept" recommendation.

Reviewer #1: All comments have been addressed

Reviewer #2: All comments have been addressed

2. Is the manuscript technically sound, and do the data support the conclusions?

Reviewer #1: Yes

Reviewer #2: Yes

3. Has the statistical analysis been performed appropriately and rigorously? 

Reviewer #1: Yes

Reviewer #2: Yes

4. Have the authors made all data underlying the findings in their manuscript fully available?

Reviewer #1: Yes

Reviewer #2: Yes

5. Is the manuscript presented in an intelligible fashion and written in standard English?

Reviewer #1: Yes

Reviewer #2: Yes

6. Review Comments to the Author

Reviewer #1: The authors have incorporated the reviewers' suggestions, both regarding the potential use of tafenoquine as adherence to treatment. Kuwiat being a country with basically imported malaria to date, its health system has proven to be efficient in preventing malaria from being endemic in this country.

The discussion as well as the abstract of the paper has been improved. I agree with the changes made. I have no further suggestions.

Reviewer #2: (No Response)

7. PLOS authors have the option to publish the peer review history of their article (what does this mean?). If published, this will include your full peer review and any attached files.

Reviewer #1: No

Reviewer #2: **Yes: **Dr. Araya Gebresilassie

---

## [Editor Report · Acceptance letter]

1 Dec 2020

PONE-D-20-30612R1 

Decreasing trend of imported malaria cases but increasing influx of mixed *P. falciparum* and *P. vivax* infections in malaria-free Kuwait 

Dear Dr. Iqbal:

I'm pleased to inform you that your manuscript has been deemed suitable for publication in PLOS ONE. Congratulations! Your manuscript is now with our production department. 

Kind regards, 

on behalf of

Dr. Luzia Helena Carvalho 

Academic Editor

PLOS ONE